

# A systematic review on machine learning approaches in cerebral palsy research

Anjuman Nahar[1], Sudip Paul[1] and Manob Jyoti Saikia[2,3]

[1] Department of Biomedical Engineering, North-Eastern Hill University, Shillong, Meghalaya, India
[2] Electrical and Computer Engineering Department, University of Memphis, Memphis, TN, United States
[3] Biomedical Sensors & Systems Lab, University of Memphis, Memphis, TN, United States

Corresponding author
Manob Jyoti Saikia,
msaikia@memphis.edu

## ABSTRACT

**Background:** This review aims to explore advances in the field of cerebral palsy (CP) focusing on machine learning (ML) models. The objectives of this study is to analyze the advances in the application of ML models in the field of CP and to compare the performance of different ML algorithms in terms of their effectiveness in CP identification, classifying CP into its subtypes, prediction of abnormalities in CP, and its management. These objectives guide the review in examining how ML techniques are applied to CP and their potential impact on improving outcomes in CP research and treatment.

**Methodology:** A total of 20 studies were identified on ML for CP from 2013 to 2023. Search Engines used during the review included electronic databases like PubMed for accessing biomedical and life sciences, IEEE Xplore for technical literature in computer, Google Scholar for a broad range of academic publications, Scopus and Web of Science for multidisciplinary high impact journals. Inclusion criteria included articles containing keywords such as cerebral palsy, machine learning approaches, outcome response, identification, classification, diagnosis, and treatment prediction. Studies were included if they reported the application of ML techniques for CP patients. Peer reviewed articles from 2013 to 2023 were only included for the review. We selected full-text articles, clinical trials, randomized control trial, systematic reviews, narrative reviews, and meta-analyses published in English. Exclusion criteria for the review included studies not directly related to CP. Editorials, opinion pieces, and non-peer-reviewed articles were also excluded. To ensure the validity and reliability of the findings in this review, we thoroughly examined the study designs, focusing on the appropriateness of their methodologies and sample sizes. To synthesize and present the results, data were extracted and organized into tables for easy comparison. The results were presented through a combination of text, tables, and figures, with key findings emphasized in summary tables and relevant graphs.

**Results:** Random forest (RF) is mainly used for classifying movements and deformities due to CP. Support vector machine (SVM), decision tree (DT), RF, and K-nearest neighbors (KNN) show 100% accuracy in exercise evaluation. RF and DT show 94% accuracy in the classification of gait patterns, multilayer perceptron (MLP) shows 84% accuracy in the classification of CP children, Bayesian causal forests (BCF) have 74% accuracy in predicting the average treatment effect on various orthopedic and neurological conditions. Neural networks are 94.17% accurate in diagnosing CP using eye images. However, the studies varied significantly in their

design, sample size, and quality of data, which limits the generalizability of the findings.

**Conclusion:** Clinical data are primarily used in ML models in the CP field, accounting for almost 47%. With the rise in popularity of machine learning techniques, there has been a rise in interest in developing automated and data-driven approaches to explore the use of ML in CP.

# INTRODUCTION

Cerebral palsy (CP) is characterized by a diverse range of mobility and posture abnormalities that are permanent but not irreversible due to injury to the developing brain. Individuals may struggle with communication, behavior, vision, hearing, nutrition, pain, and sleep in addition to their issues (*Sadowska, Sarecka-Hujar & Kopyta, 2020*). In industrialized nations, CP is thought to affect 1.4 to 1.8 out of every 1,000 live births, compared to 2.95 to 3.4 out of every 1,000 live births in low- and middle-income countries. Lifetime effects of CP include decreased independence in daily living activities, play, and involvement in educational, social, and community activities (*Graham, Paget & Wimalasundera, 2019*).

The surveillance of cerebral palsy in Europe (SCPE) has given a standardized CP classification dividing them into three major groups: spastic (unilateral or bilateral spastic), dyskinetic (dystonic or choreoathetosis), and ataxic (*Sadowska, Sarecka-Hujar & Kopyta, 2020*). The most frequent impairments in children with cerebral palsy are motor impairments, speech impairments, pain, intellectual impairments, sensory abnormalities, epilepsy, and behavioral issues. The most significant problem is motor impairments primarily caused by spasticity. Aberrant motor functions cause altered movement and posture. In addition to equinus deformity and hand dysfunction, it might result in hip discomfort or dislocation (*Zhang, 2017*; *Song et al., 2022*). Strength, balance, coordination, sensory processing, and selective motor control are common difficulties for kids with cerebral palsy. Additionally, unlike children who are usually growing, they cannot learn motor patterns (*Sadowska, Sarecka-Hujar & Kopyta, 2020*; *Graham, Paget & Wimalasundera, 2019*).

CP signs and symptoms typically appear in the early periods of infancy, yet it takes an average of 2 years for CP to be diagnosed. Because infants have a higher chance of recovering from brain damage than adults, early detection and treatment are essential for people with CP. Identifying high-risk neonates, tracking neurodevelopment, and predicting CP can all be aided by neuroimaging, motor evaluation, and neurological exams. The infant brain's structural alterations can be detected using neuroimaging techniques, including magnetic resonance imaging (MRI) and cranial ultrasonography. These techniques can also be used to track lesions' progression and evaluate treatment benefits (*Sadowska, Sarecka-Hujar & Kopyta, 2020*). Although CP cannot be cured,

medication, surgery, and other interventions like physical, occupational, speech, and behavioral therapy lead to a significant functional outcome to make the child functionally independent (*Zhang, 2017*).

However, only qualified medical professionals can carry out such an evaluation. General movement evaluations carried out by medical professionals based on visual observation are frequently influenced by observer fatigue and subjective impressions. It is necessary to develop a systematic model to deliver accurate and quick prediction outcomes and provide accurate personalized care (*Song et al., 2022*).

In recent years, machine learning has become a robust tool with enormous potential in the healthcare industry. Machine learning (ML) involves a set of multivariate analytical techniques that first identify the key features or patterns in the data that most effectively distinguish between different classes in the training set. These identified features or patterns are then used to classify or predict outcomes for new data in the test set (*Al-Sowi et al., 2023*). By utilizing machine learning techniques, it is possible to identify the disease, classify CP children into various subtypes of CP so that specific treatment interventions can be tailored for them, and predict the abnormalities and their specific treatment outcomes. With machine learning algorithms, large-scale data analysis, the extraction of significant patterns, and the development of individualized models that forecast the success of a certain intervention for a given patient are all possible (*Song et al., 2022*). RF, Support Vector Machines (SVM), MLP, artificial neural networks (ANN), direct matching, virtual twins, and Bayesian causal forests (BCF) are some of the ML models that have been increasingly applied to the field of CP (*Zhang, 2017*; *Schwartz, Ries & Georgiadis, 2021*, *2022*). Machine learning revolves around the idea that computers can learn from data, recognize patterns, and make decisions with minimal human intervention. Algorithms are trained using datasets that can be labelled, unlabelled, or a mix of both. Supervised learning (SL) employs labelled data to forecast outcomes, involving techniques such as regression, which examines the relationships between dependent and independent variables, and classification, which sorts data into predefined categories. Unsupervised learning, on the other hand, involves training machines without predefined outputs, uncovering patterns, and forming data clusters through clustering and association analysis. Reinforcement learning (RL) allows machines to identify optimal behaviours through trial and error, with feedback reinforcing positive results. Common classification methods include frequent table algorithms like Zero R, One R, and decision tree; covariance matrix algorithms such as linear discriminant analysis and logistic regression; similarity measures like K-nearest neighbors (KNN); vector and margin techniques using SVM; and neural networks like convolutional neural networks (CNN), feedforward neural networks, and feedback neural networks. Ensemble algorithms including bagging, boosting (*e.g.*, AdaBoost), and random forest (RF) combine multiple models to create the best predictive models. Logistic regression (LR) is a probabilistic approach to classification that predicts binary outcomes. KNN classifies data based on proximity to other data points, while SVM are popular for data classification and prediction. Neural networks are capable of handling both regression and classification tasks simultaneously (*Alnuaimi & Albaldawi, 2024*).

Despite advancements in therapeutic approaches and medical technology, diagnosing, classifying, and managing CP remains challenging, time consuming and reliant on the expertise of medical specialists. ML's ability to efficiently handle large datasets with objectivity and speed may lead to more accurate diagnoses and personalized treatment strategies (*Al-Sowi et al., 2023*; *Ozates et al., 2023*; *Shabber, Bansal & Radha, 2023*; *Black, Kueper & Williamson, 2023*).

The objectives of this study are to examine the progress in applying ML models within the field of CP and to evaluate the effectiveness of various ML algorithms in identifying CP, classifying its subtypes, predicting abnormalities associated with CP, and managing the condition. This review systematically examines the literature, identifying the most effective ML models, and highlights areas for further research bridging the gap between CP and ML research, thereby improving the lives of individuals with CP.

The target audience for this narrative review is broad, encompassing medical professionals, clinical researchers, and data scientists involved in CP. Medical professionals can utilize insights from ML models to improve diagnosis accuracy and optimize treatment plans for children with CP. Clinical researchers can leverage the findings to guide their investigations into the causes, development, and management of CP. Data scientists can use this knowledge to create and refine predictive models in healthcare. Additionally, patients and caregivers can benefit from summaries and implications of the findings to understand the role of emerging technologies in enhancing quality of life and potential advancements in CP care.

## SURVEY METHODOLOGY

A systematic review was conducted on Machine Learning Approaches in cerebral palsy research. Figure 1 is a flow chart showing selection of publications for inclusion in this systematic review based on PRISMA (Preferred Reporting Items for Systematic Reviews and Meta-analyses) 2020 guidelines. The PRISMA checklist has been uploaded in the Supplemental File section of the article. The search engines used during the review included electronic databases like PubMed for accessing biomedical and life sciences, IEEE Xplore for technical literature in computer, Google Scholar for a broad range of academic publications, Scopus and web of science for multidisciplinary high impact journals with search queries such as "Cerebral Palsy and Machine Learning", "Machine Learning in CP identification", "Machine learning in CP diagnosis", and "Machine learning for CP treatment outcome". Inclusion criteria included articles containing keywords such as Cerebral palsy, machine learning approaches, outcome response, identification, classification, diagnosis, and treatment prediction. We selected full-text articles, clinical trials, randomized control trial, systematic reviews, narrative reviews, and meta-analyses published in English. Studies were included if they reported the application of ML techniques for CP patients. Peer reviewed articles from 2013 to 2023 were only included for the review. Each search was constructed using the Boolean operators AND and OR to optimize search criteria. Exclusion criteria for the review included studies not directly related to CP. Editorials, opinion pieces, and non-peer-reviewed articles were also excluded.

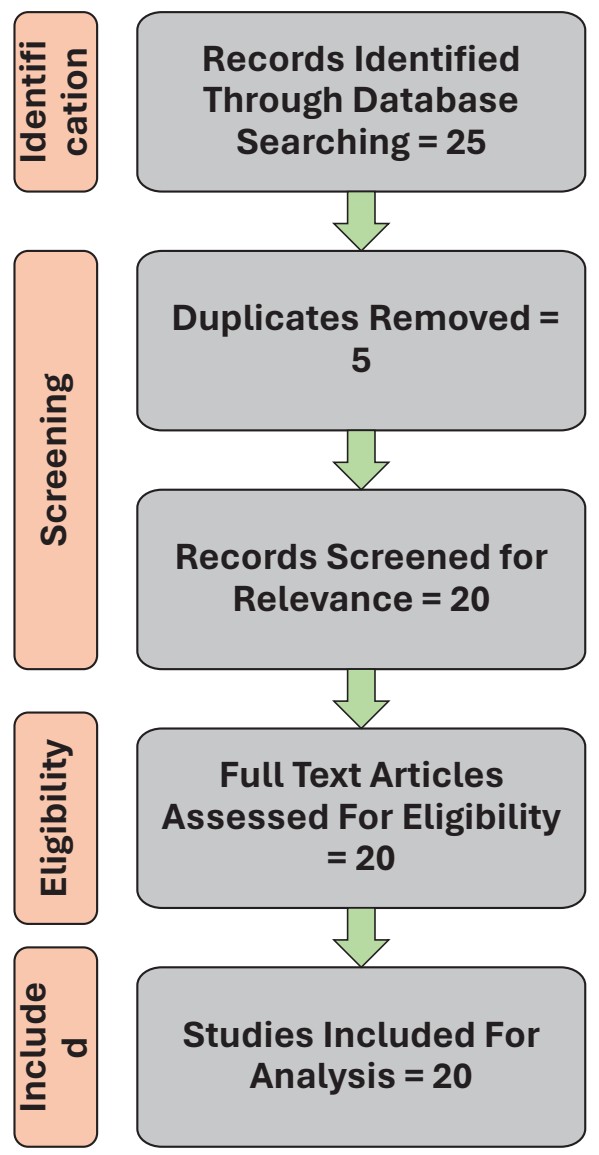

**Figure 1** Flow chart showing selection of publications for inclusion in this systematic review.

From the combination of database searches, 25 articles were identified. EndNote was used to collate all relevant articles and remove any duplicates. After screening these 25 articles, 20 article met the requirements and were used for analysis. To be included in the review, each articles was screened for title and abstract followed by full-text screening for identifying the inclusion and exclusion criteria.

By adhering to this structured approach, the review ensures a comprehensive and unbiased coverage of the literature, capturing the current state of research and advancements in the application of machine learning to cerebral palsy.

# RESULTS

After selecting the articles matching the needs of this study, we obtained 20 articles that discussed cerebral palsy with machine learning. Table 1 is the result of the articles discussed in this study. Of the 20 research articles included, we found that various ML models are useful in classification of cerebral palsy children into its subtypes, identification of abnormalities and detection of CP and prediction of treatment outcome of CP. These ML models were used for classification of human gait patterns related to cerebral palsy (CP), joint movement prediction (*Al-Sowi et al., 2023*; *Shabber, Bansal & Radha, 2023*). Identification of degree of gross motor impairment (*Black, Kueper & Williamson, 2023*) and many more described in Table 1. The ability to accurately classify gait patterns allows for more personalized treatment plans. By understanding the specific gait abnormalities of individuals, clinicians can tailor interventions to address those specific issues, potentially improving mobility and quality of life for CP patients. This approach can lead to the development of targeted physical therapy and rehabilitation programs. It also provides a foundation for further research into the underlying causes of different gait abnormalities, which could inform future therapeutic strategies and improve outcomes for patients. Identification of gait deviations in cerebral palsy children (*Breiman, 2001*), predict average treatment effect of orthopedic and neurological techniques (*Alnuaimi & Albaldawi, 2024*) and categorizing kinematic data obtained from an IMU-based device while performing nine different upper extremity activities (*Novak et al., 2017*) are some of the studies done in the field of CP using ML. These studies of ML models on CP are significant as early and precise identification of gait deviations is crucial for timely intervention. Detecting these deviations early can prevent the worsening of symptoms and aid in the development of corrective measures that can be implemented while the child is still developing. This capability enhances the early diagnostic process, enabling healthcare providers to intervene sooner. It also supports ongoing monitoring of the effectiveness of interventions, allowing for adjustments in treatment plans based on real-time data. Predictive analytics provides valuable insights into which treatments are likely to be most effective for individual patients. This can lead to more efficient and effective use of medical resources and better patient outcomes. Personalized medicine becomes more achievable with the ability to predict treatment outcomes. This can reduce trial-and-error approaches, minimize unnecessary treatments, and focus efforts on methods that are statistically shown to be beneficial, thereby improving the overall efficiency of CP management.

The ability to categorize and analyze kinematic data leads to better-designed therapeutic exercises and assistive devices. It can guide the development of interventions that specifically target the most affected movements, thereby enhancing the patient's ability to perform daily tasks independently.

To achieve the benefits of the bridge between ML and CP, research articles have been collected and are grouped based on the ML models, which is presented in Table 2. Different performance evaluation metrics were used across the studies that we have reviewed to assess the effectiveness of the machine learning models, and commonly reported metrics included accuracy, sensitivity, specificity, area under the curve (AUC),

**Table 1 List of articles selected for analysis in this study.**

| Sl. No | Study | Year | Algorithm applied | Type of data | Objectives | Outcome | Accuracy |
|---|---|---|---|---|---|---|---|
| Studies that are based on classification | | | | | | | |
| 1. | Al-Sowi et al. (2023) | 2023 | CNN, self-normalizing neural networks, RF, DT | Gait data | Classification of human gait patterns related to cerebral palsy (CP). | Random forests and decision trees produced better outcomes and focused on clinically relevant regions. | 93.40% |
| 2. | Ozates et al. (2023) | 2023 | TT-PredictMed | Clinical data | Using a predictive model to predict postural instability in children with cerebral palsy | The predictive model's average accuracy was 82%, consistent with current research on applying ML models in the clinical setting. | 82% |
| 3. | Speiser, Durkalski & Lee (2015) | 2023 | MLP, Naïve Bayes (NB), Random tree (RT), and SVM | Clinical data | Classification of cerebral palsy children using machine learning | Multilayer perceptron (MLP) accurately classifies cerebral palsy. | 84% |
| 4. | Almuaimi & Albaldawi (2024) | 2022 | Direct matching, virtual twins, and Bayesian causal forests | Clinical data | To predict the average treatment effect on the treated for 13 common orthopedic and neurological treatments using well-established causal inference approaches. | BCF performed remarkably well and offered more precise and accurate treatment predictions. | 74% |
| 5. | Novak et al. (2017) | 2022 | RF, LinearSVC, KNN, and MLP | IMU data | To identify machine learning models for categorizing kinematic data obtained from an IMU-based device while performing nine different upper extremity activities. | The RF models had the most excellent accuracy for categorizing kinematic data obtained from an IMU-based device while performing nine different upper extremity activities. | 98.60% |
| 6. | Ihlen et al. (2019) | 2022 | SVM, DT, RF, KNN | Clinical data | To identify an automated limb exercise evaluation mechanism based on machine learning techniques. | All the models achieved 100% accuracy in classifying whether an exercise was executed well. | 100% |
| 7. | Pinto-Martin et al. (1995) | 2020 | DT, RF, SVM, linear discriminant analysis, and MLP | IMU data | To identify wearable technology and machine learning models to create a clinically helpful index while also giving rehabilitation patients a chance to track their level of spasticity even in settings outside healthcare facilities. | RF performed well among all the models. | 95.40% |
| 8. | Han et al. (2002) | 2019 | SVM, RF, and NN | Eye images | To diagnose cerebral palsy using eye images. | Neural Network (NN) is found to be the most accurate. | 94.17% |
| 9. | McIntyre et al. (2011) | 2019 | DT, SVM, and RF | IMU data | To identify and test machine learning models for automatically detecting and categorizing Physical Activity types in CP children who utilize ambulation assistance. | RF model was the most accurate in automatically detecting and categorizing Physical Activity types in CP children who utilize ambulation assistance. | 74% |

(Continued)

| Sl. No | Study | Year | Algorithm applied | Type of data | Objectives | Outcome | Accuracy |
|---|---|---|---|---|---|---|---|
| 10. | Zhang & Ma (2019) | 2018 | RF, SVM, and BDT | Clinical data | To identify and evaluate ML models for automatically identifying physical activity in ambulant CP children. | SVM provided significantly better classification accuracy. | 82–89% |
| 11. | Schwartz, Ries & Georgiadis (2021) | 2017 | SVM, NN, AdaBoosted decision tree, and dynamic time warping | Clinical data | To identify the quality of exercises prescribed to CP | AdaBoosted decision tree performed the best with high classification accuracies. | 90–94% |
| 12. | Griffiths et al. (2010) | 2017 | RF, DT, and KNN | Gait data | To classify foot diseases and find the accuracy of disease detection and diagnosis. | 100% for Random Forest (RF), Decision Tree (DT), and k-nearest neighbors (KNN), and 98% for Logistic Regression. | 98–100% |
| 13. | Novak et al. (2020) | 2016 | SVM, single and double-NN, boosted decision trees, and dynamic time warping (DTW) | Clinical data | To predict the quality of an exercise and judge if it was "good" or "bad". | The Ada Boosted tree fared the best, proving the viability of exercise quality evaluation. | 94.68% |

**Studies based on regression**

| Sl. No | Study | Year | Algorithm applied | Type of data | Objectives | Outcome | Accuracy |
|---|---|---|---|---|---|---|---|
| 1. | Shabber, Bansal & Radha (2023) | 2023 | CNN models | Gait data | Joint moment prediction from kinematics | Joint movement kinematics may be predicted by the CNN model in cerebral palsy children. | nRMSE = 18.02–13.58% |
| 2. | Black, Kueper & Williamson (2023) | 2023 | Feed-forward neural net (FNN), RF, SVM, extreme gradient boosting (XG Boost) | Clinical data | Identification of degree of gross motor impairment in kids and teenagers with CP | The most accurate algorithm was the random forest one. | nRMSE = 10.1% |
| 3. | Breiman (2001) | 2023 | Adaptive boosting regression, KNN, DT, regression, random forest regression, and gradient boost regression | Gait data | Identification of gait deviations in cerebral palsy children | The gradient-boosting regression model produced the best outcome. | 79% |
| 4. | Tataranno et al. (2021) | 2016 | SVR, ADA Boost regressor, RF, linear regression, and Bayesian regression | Clinical data and EEG data | To predict the results of treatment plans for children with hand function impairment among cerebral palsy | ML prediction is more accurate than hand capacity test predictions. | All have low RMSE than Clinical tests |

**Table 2 Machine learning methods used in this study.**

| Models | Citation |
|---|---|
| DT | *Schwartz, Ries & Georgiadis (2021)*, *Al-Sowi et al. (2023)*, *Ihlen et al. (2019)*, *Griffiths et al. (2010)*, *Novak et al. (2020)*. |
| RF | *Al-Sowi et al. (2023)*, *Black, Kueper & Williamson (2023)*, *Novak et al. (2017)*, *Ihlen et al. (2019)*, *Pinto-Martin et al. (1995)*, *McIntyre et al. (2011)*, *Griffiths et al. (2010)*, *Morbidoni et al. (2021)*, *Baker & Kandasamy (2023)*, *Fan et al. (2018)*. |
| SVM | *Ihlen et al. (2019)*, *Zhang & Ma (2019)*. |
| GBR | *Breiman (2001)*, *Fan et al. (2018)*. |
| KNN | *Ihlen et al. (2019)*, *Griffiths et al. (2010)*, *Fan et al. (2018)*. |
| MLP | *Speiser, Durkalski & Lee (2015)*, *Morbidoni et al. (2021)* |
| ANN | *Suo (2023)*, *Fan et al. (2018)* |
| CNN | *Shabber, Bansal & Radha (2023)* |
| BCF | *Alnuaimi & Albaldawi (2024)* |
| BR | *Tataranno et al. (2021)* |
| LR | *Afifi* |
| Ensemble model | *de Oliveira* |

precision, and recall (*Schwartz, Ries & Georgiadis, 2021*; *Alnuaimi & Albaldawi, 2024*; *Al-Sowi et al., 2023*; *Ihlen et al., 2019*; *Zhang & Ma, 2019*). These metrics allow for a detailed comparative analysis of each model's efficacy in various applications related to CP. The studies are grouped into classification model and regression models. By understanding which models perform best in specific scenarios, researchers can focus on optimizing these models further, tailoring them to better meet the needs of CP research. Classification involves predicting a discrete label or category for a given input. It is used when the output variable is categorical, meaning it can take on a limited number of possible values or classes. The goal is to assign the input data to one of these predefined categories. Common algorithms used in case of classification metrics are logistic regression, SVM, decision trees, random forest, KNN and neural networks (*e.g.*, convolutional neural networks for image classification). Regression involves predicting a continuous numerical value for a given input. It is used when the output variable is a real-valued number, meaning it can take on any value within a range. The goal is to model the relationship between the input variables and the continuous output variable. Common algorithms used for regression are linear regression, polynomial regression, support vector regression (SVR), decision trees and random forest (when used for regression), gradient boosting machines (GBM), neural networks (*e.g.*, deep learning models for complex regression tasks) (*Alnuaimi & Albaldawi, 2024*).

A comparison of the machine learning approaches may reveal variations in predictive performance and feature selection. While specific models demonstrated high accuracy and robustness, others exhibited limitations in terms of generalizability and interpretability. For example, RF is a collection of classification and regression trees. It is an ensemble method that aggregates the predictions of multiple decision trees to improve accuracy and control overfitting (*Breiman, 2001*). Though DT are easy to use they often produce poor

**Table 3 Types of data used by ML models in cerebral palsy research** (*Schwartz, Ries & Georgiadis, 2021*; *Schwartz, Ries & Georgiadis, 2022*; *Alnuaimi & Albaldawi, 2024*; *Al-Sowi et al., 2023*; *Ozates et al., 2023*; *Shabber, Bansal & Radha, 2023*; *Black, Kueper & Williamson, 2023*; *Breiman, 2001*; *Speiser, Durkalski & Lee, 2015*; *Novak et al., 2017*; *Ihlen et al., 2019*; *Pinto-Martin et al., 1995*; *Han et al., 2002*; *McIntyre et al., 2011*; *Zhang & Ma, 2019*; *Griffiths et al., 2010*; *Novak et al., 2020*; *Tataranno et al., 2021*).

| Type of data | ML models |
|---|---|
| Clinical data | DT, RF, MLP, BCF, SVM, KNN, BR, LR |
| Gait data | RF, GBR, CNN |
| IMU data | RF |
| Image video data | NN |

accuracy. In a random forest approach, numerous classification and regression trees are built using randomly chosen subsets of the training data and randomly selected subsets of predictor variables to model the outcomes. The predictions from each individual tree are then aggregated to provide a final prediction for each observation. Thus, it provides better result than a single decision tree. It can also handle a large data set with numerous variables. However when interpretation is considered, while techniques like random forest are robust and accurate, the aggregation of multiple decision trees complicates the interpretation of individual predictions (*Speiser, Durkalski & Lee, 2015*). This necessitates research and innovative solutions to fully leverage their potential in practical applications.

After grouping the articles as per the models, we have found those ML models which are used for identifying risk factors, classification of patterns in CP children, prediction of treatment outcome, and diagnosis of CP (Table 2). Identifying risk factors associated with CP early in a child's development allows for timely intervention, potentially mitigating the severity of symptoms and improving long-term outcomes. Classifying patterns in CP children enables healthcare providers to tailor therapies and interventions to the specific needs of each child. For instance, different gait patterns can be addressed with specific physical therapy techniques. ML models predicting treatment outcomes facilitate personalized medicine, where treatments are customized based on the predicted response of each patient. This leads to higher efficacy and reduced side effects. Automated diagnostic tools can identify CP in its early stages, allowing for earlier intervention which can significantly improve developmental outcomes.

Following this, Table 3 briefly provides the type of data used for corresponding ML models in the field of CP. Commonly used data in these model's included clinical assessments (*Ozates et al., 2023*), demographic characteristics (*Speiser, Durkalski & Lee, 2015*), neuroimaging data (*Breiman, 2001*), genetic markers (*Ozates et al., 2023*) *etc*. Figure 2 shows the distribution of data sets, and their corresponding ML models used in CP research. While evaluating the performance of used machine learning techniques in cerebral palsy in Table 2, we have compared the accuracy measures of used algorithms to select the best one for future use. Figure 3 demonstrates the use of machine learning models that have been in use for cerebral palsy research.

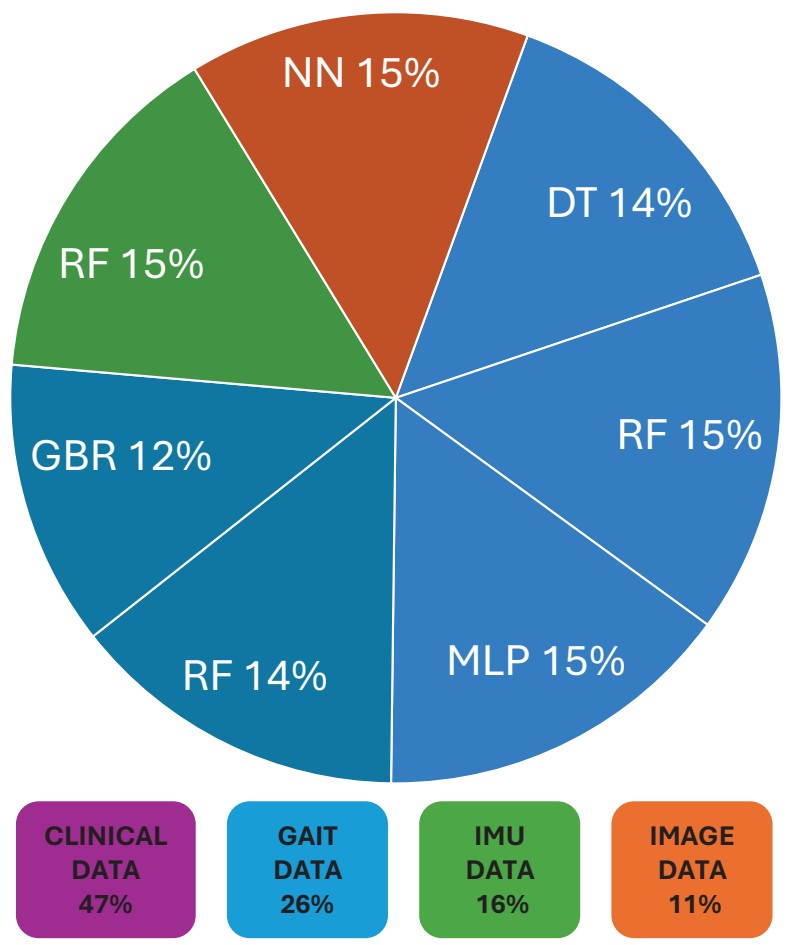

**Figure 2 Distribution of data sets and their corresponding ML models used in cerebral palsy research.**

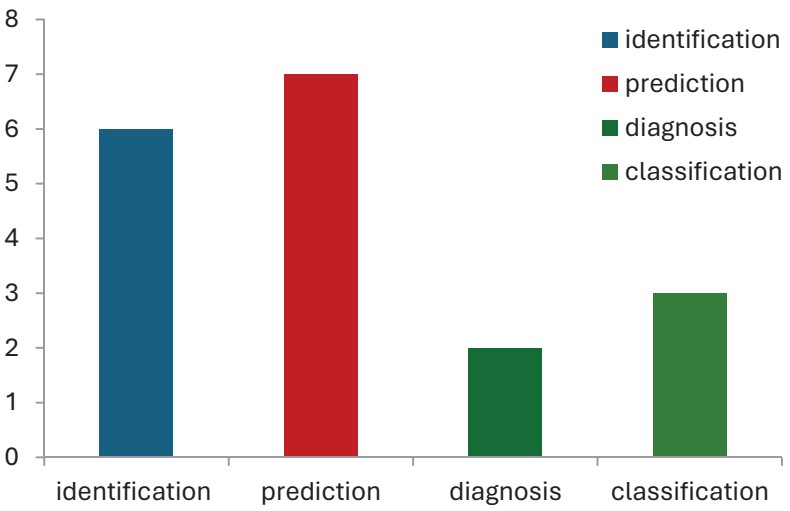

**Figure 3 Use of machine learning models in cerebral palsy.**

**Table 4 List of ML models used for identification and diagnosis of CP.**

| Study | Year | Algorithm applied | Objectives | Outcome |
|---|---|---|---|---|
| *Zhang & Ma (2019)* | 2018 | RF, SVM, and binary decision tree (BDT) | To establish and evaluate ML models for automatically identifying physical activity in ambulant CP children. | SVM provided significantly better classification accuracy. |
| *McIntyre et al. (2011)* | 2019 | DT, SVM RF | To identify and test machine learning models for automatically detecting and categorizing physical activity types in CP children who utilize ambulation assistance. | RF model was the most accurate in automatically detecting and categorizing physical activity types in CP children who utilize ambulation assistance. |
| *Black, Kueper & Williamson (2023)* | 2023 | Feed-forward neural net (FNN) Random forest (RF) Support vector machine (SVM) Extreme gradient boosting (XG Boost) | To identify the accuracy of measuring the level of gross motor impairment in children and adolescents with CP. | The random forest algorithm proved to be the most accurate. |
| *Breiman (2001)* | 2023 | Adaptive boosting regression, K-nearest neighbor, decision tree regression, random forest regression, and gradient boost regression. | Identification of gait deviations in cerebral palsy children. | The best result was obtained using the gradient-boosting regression model. |
| *Novak et al. (2017)* | 2022 | Random forest (RF), LinearSVC, k-Nearest neighbors (kNN), and multilayer perceptron (MLP) | To identify machine learning models for categorizing kinematic data obtained from an IMU-based device while performing nine different upper extremity activities. | The RF models had the most excellent accuracy for categorizing kinematic data obtained from an IMU-based device while performing nine different upper extremity activities. |
| *Ihlen et al. (2019)* | 2022 | Support vector machines, decision trees, random forests, and k-nearest neighbors | To identify an automated limb exercise evaluation mechanism based on machine learning techniques. | All the models achieved 100% accuracy in classifying whether an exercise was executed well. |
| *Pinto-Martin et al. (1995)* | 2020 | Decision tree, random forests (RFs), support vector machine, linear discriminant analysis, and multilayer perceptrons | To identify wearable technology and machine learning models to create a clinically helpful index while also giving rehabilitation patients a chance to track their level of spasticity even in settings outside of healthcare facilities. | RF performed well among all the models. |
| *Schwartz, Ries & Georgiadis (2021)* | 2017 | SVM, neural networks, Ada Boosted decision tree, and dynamic time warping | To identify the quality of exercises prescribed to CP. | AdaBoosted decision tree performed the best with high classification accuracies. |
| *Morbidoni et al. (2021)* | 2018 | logistic regression and random forest | Comparison of ML models to identify cases of cerebral palsy from unidentified cases. | RF models are a reliable and affordable method to locate cerebral palsy instances that may not have been previously identified. |
| *Baker & Kandasamy (2023)* | 2020 | RF | To identify cerebral hemorrhage in preterm infants using the RF model. | It has good predictability to identify cerebral hemorrhage in preterm infants. |

## Identify abnormalities in CP children and diagnosis of CP using ML Models

The primary diagnostic method for CP identification still relies on traditional clinical assessment components like delayed motor milestones, asymmetry of movement, or abnormal muscle tone, scales like The General Movements Assessment, The

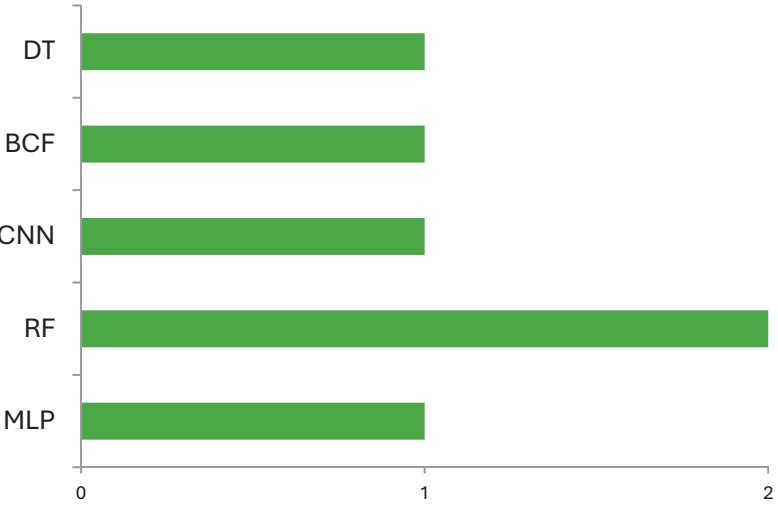

**Figure 4** Machine learning models used for identification and diagnosis of cerebral palsy.

Hammersmith Infant Neurological Examination, and neurological data like MRI (*Novak et al., 2017*) which are time consuming, expensive and needs specialist's observation. However, as machine learning techniques have become more prevalent, there has been an increase in interest in creating automated and data-driven methods to recognize and diagnose CP, improving accuracy and effectiveness (*Ihlen et al., 2019*). For instance, ML approaches have made it possible to analyze recorded abnormal movement data in CP children like lack of smoothness and fluidity, often appearing stiff and jerky and typically small, variable, and continuous movements whose presence make a strong possibility of CP diagnosis. A list of some of the ML models used for identifying symptoms and diagnosing CP is given in Table 4.

Table 4 explains which data can be used by ML models for identification of various abnormalities of CP children. Most of the studies use variables such as physical activities in neonates, gross motor activities of ambulant, gait data, image data for diagnosis of CP and identification of various abnormalities and type of management that can be prescribed for CP children. This includes gait deviations, quality exercises prescribed for CP children, causes that led to CP. On comparison Fig. 4 demonstrates the machine learning models used for identification and diagnosis of cerebral palsy. It shows that the Random Forest model is the most accurate ML model for identifying movements and disease-causing factors in CP children. For identification of risk factors multivariate logistic regression has often been used (*Al-Sowi et al., 2023*; *Pinto-Martin et al., 1995*). The variables used by these studies for identification of risk factors are premature birth, low birth weight, severe birth asphyxia, preterm rupture of membrane, abnormal cranial ultrasound, or structural MRI imaging findings, intraventricular hemorrhage, PVL, neonatal sepsis, hypoxia-ischemic encephalopathy, hypoglycemia, neonatal jaundice, *etc.* (*Pinto-Martin et al., 1995*; *Han et al., 2002*). These risk factors are useful to understand the causes of CP, identify high-risk infants, and aid in the diagnosis of CP. Early brain injury can be avoided by preventing their underlying cause and managing with hypothermia, rubella vaccination, anti-D

**Table 5 List of ML models used for the classification of cerebral palsy.**

| Study | Year | Algorithm applied | Objectives | Outcome |
|---|---|---|---|---|
| *Al-Sowi et al. (2023)* | 2023 | Convolutional neural networks, self-normalizing neural networks, random forests, and decision trees | Classification of human gait patterns related to cerebral palsy (CP). | Random forests and decision trees achieve better results and focus more on clinically relevant regions compared. |
| *Speiser, Durkalski & Lee (2015)* | 2023 | K-Star multilayer perceptron (MLP) Naïve Bayes (NB) random tree (RT) and support vector machine (SVM) | Classification of cerebral palsy children using machine learning. | Multilayer perceptron (MLP) accurately classifies cerebral palsy. |
| *Griffiths et al. (2010)* | 2017 | Random forest (RF), decision tree (DT), and k-nearest neighbors (KNN), | To classify foot diseases and find the accuracy of disease detection and diagnosis. | Random forest (RF), Decision Tree (DT), and k-nearest neighbors (KNN) and 98% for Logistic Regression. |
| *Suo (2023)* | 2019 | Artificial neural network (ANN), discriminant Analysis, naive Bayes, decision tree, *k*-nearest neighbors (KNN), support vector machine (SVM), and random forest | To assess the effectiveness of machine learning techniques for categorizing the gait patterns of CP children. | The most accurate prediction method is ANN. |
| *Fan et al. (2018)* | 2018 | ANN, MLP, SVM, KNN, DT classifiers, RF, GBM | To classify fetuses who are suffering from oxygen deprivation using ML models. | All models show good accuracy. ANN, DT classifiers, RF, and GBM, have demonstrated good accuracies using various variables. |

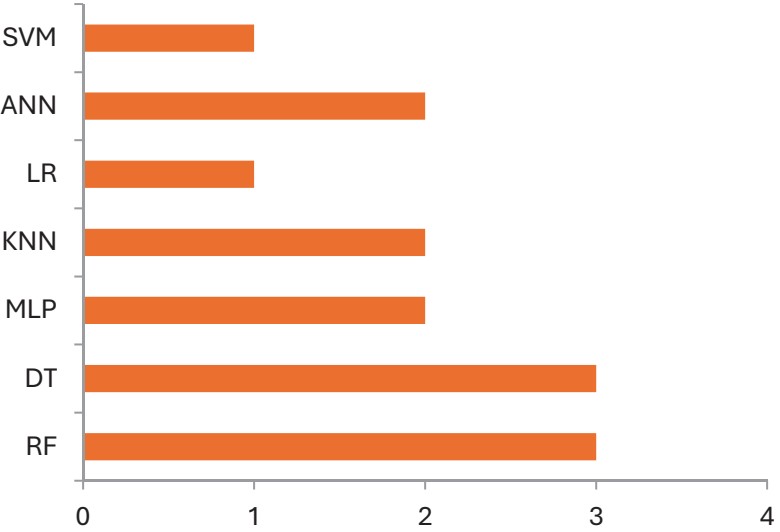

**Figure 5 Machine learning models used for classification in cerebral palsy cases.**

vaccination, and preventing methylmercury contamination (*Al-Sowi et al., 2023*; *McIntyre et al., 2011*). Since, currently, there is no cure for CP, its prevention is critical to reduce the prevalence of CP and save children from CP and its life-long disabilities. Multivariate analysis may help identify significant risk factors and prevent CP (*Al-Sowi et al., 2023*).

## Classification of CP children into subtypes of CP using ML models
Classication of children with CP is challenging for clinicians due to the similarity in their clinical presentations. It is essential to correctly classify CP subtypes to comprehend

**Table 6 List of machine learning models used for prediction in cerebral palsy cases.**

| Study | Year | Algorithm applied | Objectives | Outcome |
|---|---|---|---|---|
| *Morbidoni et al. (2021)* | 2021 | Support vector machine (SVM), random forest (RF), K-nearest neighbors (KNN), and multilayer perceptron (MLP) | To predict locomotion events n hemiplegic CP | MLP and RF have good prediction accuracies |
| *Ozates et al. (2023)* | 2023 | TT-PredictMed | Using a Predictive model to predict postural instability in children with cerebral palsy | The accuracy of the predictive model was 82% on average, which is in line with recent studies on using machine learning models in the clinical field. |
| *Shabber, Bansal & Radha (2023)* | 2023 | CNN models | Joint moment prediction from kinematics | Joint movement kinematics may be predicted b the CNN model in cerebral palsy children. |
| *Alnuaimi & Albaldawi (2024)* | 2022 | Direct matching, virtual twins, and Bayesian causal forests | To predict the average treatment effect on the treated 13 common orthopedic and neurological treatments using well-establish causal inference approaches. | BCF performed exceptionally well and provided more accurate and precise treatment predictions than other causal inference methods |
| *Novak et al. (2020)* | 2016 | Support vector machines (SVM), single and double-layered neural networks (NN), boosted decision trees, and dynamic time warping (DTW) | To predict the quality of an exercise and judge if it was "good" or "bad". | The Ada Boosted tree fared the best proving the viability of exercise quality evaluation. |
| *Tataranno et al. (2021)* | 2016 | Support vector regressor, ADA boost regressor, random forest regressor, linear regression and Bayesian regression | To predict the results of treatment plans for children with hand function impairment among cerebral palsy. | Bayesian regression has good accuracy in predicting treatment plans for children with CP. |
| *Afifi* | 2021 | Random forest (RF), logistic regression | To predict CP in very preterm infants. | Both are comparable and predictable in CP prediction. |
| *Krechowicz, Deniziak & Kaczmarski (2023)* | 2023 | Adaptive boosting regression, K-nearest neighbor, decision tree regression, random forest regression, and gradient boost regression. | To predict gait deviation in cerebral palsy children | Gradient boost regression shows better result |
| *de Oliveira* | 2023 | Ensemble models, support vector machines, and artificial neural networks | To identify a machine learning model to predict patients' motor functions after therapy in rare disorders like cerebral palsy. | Ensemble models |

underlying mechanisms, forecast outcomes, and create individualized therapies (*Al-Sowi et al., 2023*). Automatically identifying CP subtypes based on various data sources using machine learning algorithms has shown remarkable results, enabling more accurate diagnosis and individualized treatment plans (*Zhang & Ma, 2019*). For instance a study (*Han et al., 2002*) differentiated CP types by examining the T2 MRI images of children as spastic or dyskinetic according to severity of injury. However Multivariate logistic regression further categorised the injuries acording to the area of lesion into spastic and dyskinetic CP (*Griffiths et al., 2010*). Table 5 presents some of the studies done in the classification field in cerebral palsy children. In Fig. 5 which demonstrates the machine learning models used for classification in cerebral palsy cases, we can see that decision trees and random forests are accurate classifying models in cases of CP.

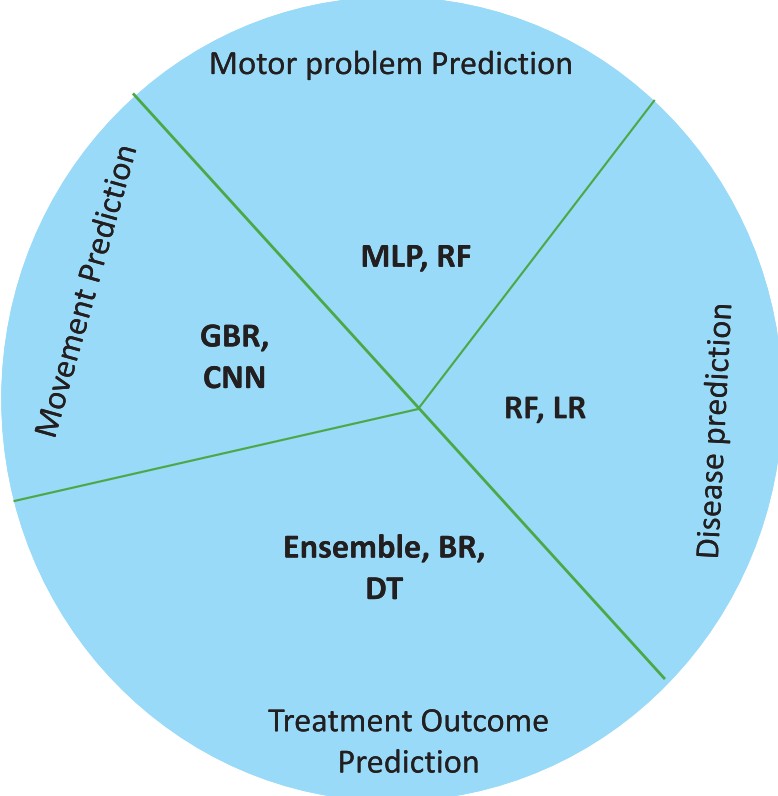

**Figure 6 Use of ML models for prediction in cerebral palsy cases.**

## Treatment response predictions in CP children using ML models

Healthcare professionals need to foresee probable complications, anticipate the course of the condition, and develop individualized treatment plans (*Novak et al., 2020*). Thus, in CP, machine learning has become a potent tool for predictive modeling, utilizing vast amounts of data and advanced algorithms to produce precise prognostic insights (*Tataranno et al., 2021*). This section focuses on how machine learning methods have been used to forecast various CP-related outcomes, from treatment outcomes and disease progression to functional products and adaptive solutions. Machine learning models have been found to be cost-effective and accurate prediction tools in neurological cases (*Ihlen et al., 2019*). For instance, due to the neuromuscular involvement in hemiplegic CP, it is strongly encouraged to assess the recruitment of muscles using myoelectric-signal analysis. In this case, additional, expensive, and complex attributes are needed to identify the gait components. In such cases ML is successfully used to interpret gait data, to detect gait events in healthy subjects. Future developments in this area is suggested if involving ML models may improve performances of CP children (*Morbidoni et al., 2021*). Table 6 lists the ML models used for predicting disease and its features, followed by treatment responses in CP. Figure 6 demonstrates the use of ML models for prediction in cerebral palsy cases. Prediction models predict the disease, motor problems due to the disease, moments, and treatment responses. The ML models used for disease prediction are RF and

LR; for movement prediction, GBR and CNN; for motor problems, predictions are MLP and RF; for treatment prediction, are Ensemble, BR, and DT models.

## DISCUSSION

This report analyzed numerous studies and research initiatives that showed which machine learning algorithms might improve the accuracy and effectiveness of CP identification and management. The following discussion illustrates the difficulties and potential possibilities in this developing field while summarizing the main findings, discussing the implications, and highlighting the challenges.

Machine learning algorithms have demonstrated the ability to predict motor development delays and identify newborns more likely to acquire cerebral palsy (*Zhang, 2017*; *Baker & Kandasamy, 2023*). Thanks to these predictive models, the ability to intervene early facilitates early intervention therapy and lessens the condition's potential effects. However, successfully implementing these models in clinical settings requires rigorous validation and the establishment of clear guidelines for their interpretation and integration (*Black, Kueper & Williamson, 2023*; *Breiman, 2001*; *Speiser, Durkalski & Lee, 2015*; *Novak et al., 2017*; *Ihlen et al., 2019*; *Pinto-Martin et al., 1995*; *Han et al., 2002*; *McIntyre et al., 2011*; *Zhang & Ma, 2019*). By enabling personalized therapeutic recommendations, machine learning has the potential to revolutionize cerebral palsy treatment planning. Predictive models can suggest the best treatments for each patient by examining their unique patient data, such as clinical history, neuroimaging findings, and treatment responses. This individualized strategy optimizes the use of healthcare resources while simultaneously improving treatment outcomes. However, when implementing personalized therapy recommendations, ethical issues, including data protection and patient permission, are essential (*Alnuaimi & Albaldawi, 2024*; *Ozates et al., 2023*; *Shabber, Bansal & Radha, 2023*; *Breiman, 2001*; *Novak et al., 2020*; *Tataranno et al., 2021*).

As mentioned in the literature, the challenges and limitations associated with using ML in the context of CP mainly revolve around the need for high-quality datasets for model training. ML algorithms heavily rely on reliable datasets. However, gathering well-curated datasets for CP can be challenging. The availability of representative data, such as records, imaging data, and longitudinal patient information, is often limited. This limitation can potentially impact the performance and generalizability of ML models with health data (*Suo, 2023*).

According to the finding of this study the RF is the most accurate ML model for identifying movements and disease-causing factors. Decision trees and random forests are accurate classifying models in cases of CP. The ML models used for disease prediction are RF and LR; for movement prediction, GBR and CNN; for motor problems, predictions are MLP and RF; for treatment prediction, are Ensemble, BR, and DT models. The data sets mostly used by ML models in CP research are clinical data which accounts for 47%, followed by gait data accounting for 26%. DT, MLP and RF are mostly used on clinical data. It was found that ML models in cerebral palsy have been used to identify the risk factors, predict the disease, diagnosis, and treatment response and Fig. 4 explains that ML models in CP are mainly used for prediction. Prediction here indicates treatment

response outcomes, disease progression, motor problems prediction and prediction of developing CP in early stages. Our study is at per with *Zhang (2017)* who stated that multivariate analytic methods are useful for identifying risk factors, detecting CP, predicting CP through movement assessment, and evaluating outcomes. ML approaches enable the automatic identification of movement impairments in high-risk infants. Additionally, multivariate outcome studies have identified predictors for surgical treatment outcomes. This study is in consistence with *Fan et al. (2018)* which interprets that Data-driven approaches like RF can identify the most informative predictors, helping to reliably detect potential unrecorded cases of cerebral palsy or other complex medical conditions in primary care databases. Future research in this area is suggested to Develop models focused on early detection of CP in infants to enable timely interventions that could improve long-term outcomes, Create ML models that tailor treatment plans based on individual patient data, enhancing the effectiveness of interventions, Apply ML models to diverse populations to ensure the generalizability and robustness of the findings across different demographic and geographic groups and Implement and test ML models in real-world clinical settings to evaluate their practical utility and impact on patient care.

## CONCLUSIONS

This in-depth analysis of various machine learning (ML) models highlights their substantial potential in enhancing the detection, diagnosis, and treatment of cerebral palsy (CP). The studies reviewed demonstrate the adaptability of ML models in addressing numerous aspects of CP management, ranging from early abnormalities identification to the development of personalized treatment plans. Findings suggest that RF is often accurate in identifying movements and disease-causing factors in CP children. Both DT and RF models prove to be effective in classifying CP subtypes, which is crucial for crafting targeted interventions. Clinical data, comprising 47% of the datasets, emerges as the most frequently used data type in ML models for CP research, followed by gait data at 26%. Models such as DT, MLP, and RF are predominantly applied to clinical data, showcasing their versatility in handling diverse data types. ML models are mainly employed for predictive purposes in CP, encompassing treatment response outcomes, disease progression, and early-stage CP prediction. Through predictive analytics, personalized medicine is facilitated, thereby optimizing treatment efficacy and resource utilization by tailoring interventions to individual patient profiles. However, the effectiveness of ML models is largely contingent upon the quality of the datasets used. Acquiring well-curated datasets for CP poses a significant challenge, potentially affecting model performance and generalizability. The limited availability of comprehensive records, imaging data, and longitudinal patient information can restrict the applicability of these models across diverse clinical settings. Moreover, the implementation of personalized therapeutic recommendations *via* ML models raises ethical issues, particularly concerning data protection and patient consent. Addressing these concerns is essential to ensure the responsible deployment of ML in clinical practice. Developing ML models that customize treatment plans based on individual patient data can enhance intervention effectiveness, minimizing trial-and-error approaches and improving patient outcomes. To ensure the

generalizability and robustness of ML models, it is crucial to apply them to diverse populations. Additionally, testing and integrating ML models in real-world clinical settings is vital for evaluating their practical utility and impact on patient care. Such steps are necessary to seamlessly incorporate these advanced technologies into routine clinical practice.

### Funding
This research and the APC were funded by the Biomedical Sensors & Systems Lab. The funders had no role in study design, data collection and analysis, decision to publish, or preparation of the manuscript.

### Grant Disclosures
The following grant information was disclosed by the authors:
Biomedical Sensors & Systems Lab.

### Competing Interests
The authors declare that they have no competing interests.

### Author Contributions
- Anjuman Nahar conceived and designed the experiments, performed the experiments, analyzed the data, prepared figures and/or tables, and approved the final draft.
- Sudip Paul conceived and designed the experiments, performed the experiments, prepared figures and/or tables, and approved the final draft.
- Manob Jyoti Saikia conceived and designed the experiments, authored or reviewed drafts of the article, and approved the final draft.

### Data Availability
This is a literature review.

### Supplemental Information
Supplemental information for this article can be found online at http://dx.doi.org/10.7717/peerj.18270#supplemental-information.

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
