# Peer review of "A systematic review on machine learning approaches in cerebral palsy research"

_PeerJ, doi:10.7717/peerj.18270_

## Round 0.1 · original submission · Major Revisions

The reviewers report indicated that this review article has good content and believed that the information will be useful to readers. However, major changes need to be made in the presentation of the material. Please write a clear point by point response to all comments made by the reviewers, and revise the manuscript accordingly.

Reviewer 1 ·

Basic reporting

The authors provided a narrative review on application of ML on CP research. Sentence writing quality is not up the marks. There are multiple flows:
In abstract section do not give abbreviation before elaborating the terms. Citations are not in correct position (line 39 - 44). CP abbreviation has been used for cerebral palsy as well as clinical practice (line 88). Repeated information about the usage of ML in CP assessment is written in the introduction section. Throughout the paper the most significant thing is lack of citations, whether it is the result section, discussion section whatever be. Many statements require citation, which is not given at all. In a narrative review like this paper, citation is an important matter establish the fact. “Classification using ML Models” and “Predictions using ML Models”: these two heading does not refer two different things to me. Just by specifying the fact that other studies conveys does not signify any important contribution.

Experimental design

The authors provided a narrative review on application of ML on CP research. Sentence writing quality is not up the marks. There are multiple flows:
In abstract section do not give abbreviation before elaborating the terms. Citations are not in correct position (line 39 - 44). CP abbreviation has been used for cerebral palsy as well as clinical practice (line 88). Repeated information about the usage of ML in CP assessment is written in the introduction section. Throughout the paper the most significant thing is lack of citations, whether it is the result section, discussion section whatever be. Many statements require citation, which is not given at all. In a narrative review like this paper, citation is an important matter establish the fact. “Classification using ML Models” and “Predictions using ML Models”: these two heading does not refer two different things to me. Just by specifying the fact that other studies conveys does not signify any important contribution.

Validity of the findings

The authors provided a narrative review on application of ML on CP research. Sentence writing quality is not up the marks. There are multiple flows:
In abstract section do not give abbreviation before elaborating the terms. Citations are not in correct position (line 39 - 44). CP abbreviation has been used for cerebral palsy as well as clinical practice (line 88). Repeated information about the usage of ML in CP assessment is written in the introduction section. Throughout the paper the most significant thing is lack of citations, whether it is the result section, discussion section whatever be. Many statements require citation, which is not given at all. In a narrative review like this paper, citation is an important matter establish the fact. “Classification using ML Models” and “Predictions using ML Models”: these two heading does not refer two different things to me. Just by specifying the fact that other studies conveys does not signify any important contribution.

·

Basic reporting

I would like to appreciate the authors. Cerebral palsy is one of the issues encountered in clinics and is studied very seriously by researchers. Machine learning has become a phenomenon that has been studied seriously, especially in recent years. In this respect, the article has a large readership and potential to be read. However, there are major shortcomings regarding the structure of the article.

Experimental design

There are major shortcomings regarding the structure of the article. The treatment of the subject is very weak. The headlines are not full enough. Very valuable studies in the literature have been taken into account, but unfortunately the presentation is very weak and absolutely not descriptive. Evaluating the results and presenting them with tables and figures is not understandable for the reader. The order of the figures in the text and the order given at the end of the article are incompatible. Therefore, it is very difficult for the reader to follow and understand the article.

Validity of the findings

Unfortunately, the evaluation of the results, the tables and figures provided are not understandable for the reader. Moreover, many figures do not seem necessary.

Additional comments

Please be very elaborate in the writing and presentation of the article.

Reviewer 3 ·

Basic reporting

The goal of the paper was to provide a literature review of the the papers concerning machine learning methods (ML) applied to patient suffer from cerebral palsy (CP). Authors took into consideration many papers that deal with classification of the suptypes of the CP and assesing the patients condition. The paper concenrs up to date topic and suits aims and scope of the journal. In my opinion the paper is mostly interresting for medical practitioners that deals with CP patients.
In my opinion the paper is worth publishing but before that some issues needs to be improved:
- There is no clear distincion between classificaiton and regression problems that are discussed. I suggest to divide the tables with the results in such a way that it will clearly distinguish classification and regression. Classification and regression uses different metrics to evaluate models.
- I advise to organize the information about ML techniques used. For example methods such SVM, DT, ANN can be used as both regression and classification. Additionally, I suggest to provide better discussion about Ensemle methods. For example Random Forest or Gradient and Adaptive Boosted Trees can also considered as Ensemble method. What is the distinction of ANN and MLP?
- It is not clear how many papers Authors analysed (line 116: 25 papers, letter in that line "EndNote 20"?; line 124: 24 papers; also different numbers in the tables)
- Authors could include search queries that were used to find papers in the databases
- In line 33 Authors claims that the "eye images" are used. Does it mean that the image of eye are used?
- line 30 Authors claim that 100% accuracy was obtained. I suggest to validate this information since 100% accuracy is in most cases impossible to achieve.
- line 139: "While specific models demonstrated high accuracy and robustness, others exhibited limitations in terms of generalizability and interpretability." I advise to elaborate on that statement and provide concrete references.
- I suggest to extend the conclusions section by providing more summary about the results and methods.
Some minor issues:
- Some acronyms are not introduced before used (in the abstract, line 110, line 222),
- line 261 bracket not closed in citation 13
- line 324 reference 14 not properly aligned
- lines 73-76 not justified
- Fig. 5: the labels of the methods are not properly visible. Additionally, I suggest to remove the values 0.5, 1.5, etc since they are not possible in therm of the number of papers (same applies to Fig. 4)
- I suggest to unify the presentation of pie chart (e.g. Fig. 1 and 2)

Experimental design

Please see Basic reporting section.

Validity of the findings

Please see Basic reporting section.

Additional comments

Please see Basic reporting section.

---

## Round 0.2 · accepted · Accept

Thank you for doing an excellent revision to address the concerns of the reviewers.

·

Basic reporting

I would like to thank the authors for properly handling my revision request. I believe that all revisions improved the quality of the manuscript and made it suitable for publication in PeerJ.

Experimental design

No comment.

Validity of the findings

No comment.

Additional comments

No comment.